**Subject Area:**
biochemistry/structural biology

*Fusobacterium nucleatum*, pathogen, catalytic F$_1$-ATPase, structure, ATP hydrolysis, regulation

**Authors for correspondence:**
John E. Walker
e-mail: walker@mrc-mbu.cam.ac.uk
Gregory M. Cook
e-mail: gregory.cook@otago.ac.nz

[†]Present address: Medical Research Council Mitochondrial Biology Unit, Cambridge Biomedical Campus, University of Cambridge, Cambridge CB2 0XY, UK.
[‡]These authors contributed equally to the study.
[¶]Present address: Diamond Light Source, Harwell Science and Innovation Campus, Didcot OX11 0DE, UK.
[§]Present address: Research Funding Support Department, University of St Andrews, Fife KY16 9RJ, UK.

# Structure of F$_1$-ATPase from the obligate anaerobe *Fusobacterium nucleatum*

Jessica Petri[1,†], Yoshio Nakatani[1,2,‡], Martin G. Montgomery[3,‡], Scott A. Ferguson[1], David Aragão[4,¶], Andrew G. W. Leslie[5], Adam Heikal[1,2,§], John E. Walker[3] and Gregory M. Cook[1,2]

[1]Department of Microbiology and Immunology, Otago School of Medical Sciences, University of Otago, Dunedin 9054, New Zealand
[2]Maurice Wilkins Centre for Molecular Biodiscovery, The University of Auckland, Private Bag 92019, Auckland 1042, New Zealand
[3]Medical Research Council Mitochondrial Biology Unit, Cambridge Biomedical Campus, Cambridge CB2 0XY, UK
[4]Australian Synchrotron, 800 Blackburn Road, Clayton, Victoria 3168, Australia
[5]Medical Research Council Laboratory of Molecular Biology, Cambridge Biomedical Campus, Cambridge CB2 0QH, UK

DA, 0000-0002-6551-4657; GMC, 0000-0001-8349-1500

The crystal structure of the F$_1$-catalytic domain of the adenosine triphosphate (ATP) synthase has been determined from the pathogenic anaerobic bacterium *Fusobacterium nucleatum*. The enzyme can hydrolyse ATP but is partially inhibited. The structure is similar to those of the F$_1$-ATPases from *Caldalkalibacillus thermarum*, which is more strongly inhibited in ATP hydrolysis, and in *Mycobacterium smegmatis*, which has a very low ATP hydrolytic activity. The β$_E$-subunits in all three enzymes are in the conventional 'open' state, and in the case of *C. thermarum* and *M. smegmatis*, they are occupied by an ADP and phosphate (or sulfate), but in *F. nucleatum*, the occupancy by ADP appears to be partial. It is likely that the hydrolytic activity of the *F. nucleatum* enzyme is regulated by the concentration of ADP, as in mitochondria.

## 1. Introduction

The adenosine triphosphate (ATP) synthases, also known as F-ATPases or F$_1$F$_o$-ATPases, are multi-subunit enzymes found in energy-transducing membranes in mitochondria, chloroplasts and eubacteria [1,2]. They catalyse the synthesis of ATP from ADP and inorganic phosphate by using energy from a transmembrane electrochemical gradient of protons, known as the proton motive force (or pmf). Alternatively, some eubacteria generate a sodium ion motive force (or smf) to power the generation of ATP [3].

The subunits of ATP synthases are organized into membrane intrinsic and membrane extrinsic sectors [1,2]. The membrane extrinsic sector, known as F$_1$-ATPase, is the catalytic part where ATP is formed from ADP and inorganic phosphate. It can be detached experimentally from the membrane domain in an intact state and retains the ability to hydrolyse, but not to synthesize, ATP. The F$_1$-catalytic domains of bacterial ATP synthases are assemblies of five polypeptides. Three α-subunits and three β-subunits are arranged in alternation around a central stalk made from single copies of the γ- and ε-subunits, and in the intact ATP synthase, the central stalk is associated with a ring of c-subunits in the membrane domain of the complex. Together, the γε-subcomplex and the c-ring constitute the enzyme's rotor. The turning of the rotor modulates the binding properties of the three catalytic sites which lie at three of the interfaces between α- and β-subunits, taking each of them through a cycle of substrate binding, and ATP formation and release. The single δ-subunit sits on top of the α$_3$β$_3$-hexamer and, together with two identical b-subunits (or related but non-identical b- and b′-subunits in some bacterial species and chloroplasts), forms

royalsocietypublishing.org/journal/rsob    Open Biol. 9: 190066

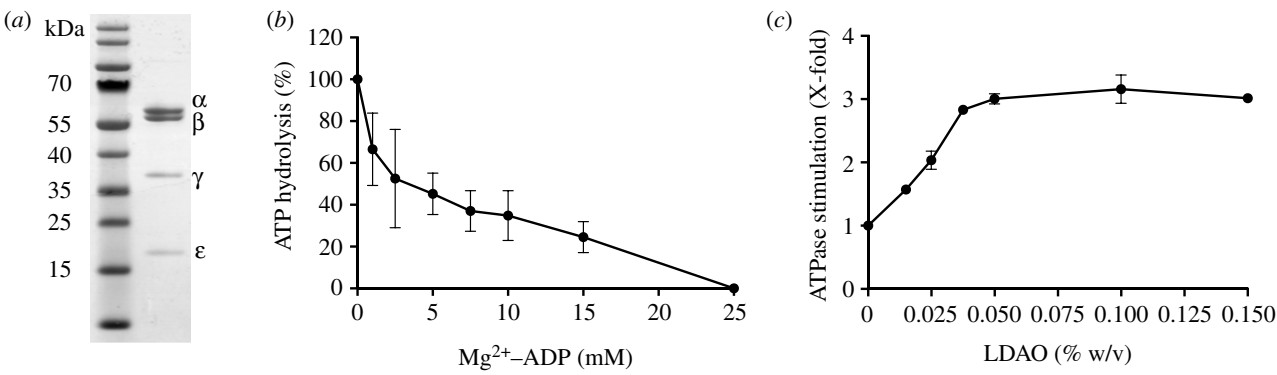

**Figure 1.** Characterization of $F_1$-ATPase from *F. nucleatum*. (*a*) The subunits (1.75 μg of enzyme) were separated by SDS—PAGE and stained with Coomassie G-250 dye. Their identities (right-hand side) were verified by mass-mapping of tryptic peptides. The positions of molecular mass markers are indicated on the left. (*b*) The effect of ADP on ATP hydrolysis at an $Mg^{2+}$ : ADP ratio of 2 : 1 was assayed by the release of inorganic phosphate. A specific activity of 3.5 U mg$^{-1}$ was set to 100%. (*c*) The effect of LDAO on ATP hydrolysis was measured with an ATP-regenerating assay. A value of 1 corresponds to a specific activity of 3.8 U mg$^{-1}$. Error bars represent the standard deviation of the mean from a biological triplicate. Where no error bars are shown, they are smaller than the diameter of the data points.

part of the stator linking the external surface of the $\alpha_3\beta_3$-domain to a single a-subunit in the membrane domain. Protons or sodium ions re-enter the bacterial cytoplasm via half channels at the interface between the c-ring and the a-subunit and deliver energy to impel the turning of the rotor.

Many ATP synthases can not only synthesize ATP, but under conditions of low pmf or smf and high intracellular ATP, they can operate in reverse, hydrolysing ATP to generate the pmf or smf, which is required for other cellular functions, such as transmembrane transport of small molecules, or, in motile bacteria, to drive the motor of the flagellum. Therefore, this hydrolytic mechanism has to be regulated in order to prevent wasteful hydrolysis of ATP. Since eubacteria live in a wide range of environments, their pmf or smf is influenced by external factors, such as pH, nutrients and oxygen tension, and can vary over a wide range [4]. Therefore, it is likely that diverse mechanisms of the regulation of bacterial ATP synthases will operate under these multifarious conditions of growth. In recent years, the need to understand these mechanisms of regulation has increased dramatically because of the increase of resistance to antibiotics of pathogenic microorganisms and the authentication of the ATP synthase of *Mycobacterium tuberculosis*, especially multidrug-resistant, extensively drug-resistant and totally resistant strains, as the target for treating tuberculosis with the drug bedaquiline [5,6]. By implication, other bacterial ATP synthases could also be developed as drug targets for treating infectious diseases, but a rational approach to the design of new drugs requires the detailed structures of the ATP synthases from the pathogens, and an understanding of how they are regulated, and how their structures, mechanisms and modes of regulation differ from those of the human enzyme. As the structure [7–26] and regulation of the closely related bovine enzyme by the inhibitor protein $IF_1$ have been well studied [9,16,27–31], the bovine enzyme provides an excellent surrogate for the human complex.

As part of this endeavour to develop bacterial ATP synthases as drug targets for treating infectious diseases, we have studied the structure and regulation of the $F_1$-catalytic domain of the ATP synthase from the opportunistic periodontal pathogen *Fusobacterium nucleatum*, which is associated with a wide range of diseases including oral infections, adverse pregnancy outcomes, gastrointestinal disorders and atherosclerosis [32]. Also, it has been linked recently to the development and progression of colorectal cancer via inhibition of anti-tumour

immune signalling pathways and the subsequent promotion of chemoresistance [33,34]. *Fusobacterium nucleatum* is an obligately anaerobic bacterium that grows by fermentative metabolism with glutamate as a substrate. A membrane-bound glutaconyl-CoA decarboxylase catalyses the decarboxylation of glutaconyl-CoA to crotonyl-CoA [35] and couples the free energy of the reaction to the transport of $Na^+$ ions across the membrane, generating an smf [36]. The ATP synthase uses this smf to drive the synthesis of ATP required for catabolic and anabolic reactions [37].

# 2. Results and discussion

## 2.1. Biochemical properties of the $F_1$-ATPase from *Fusobacterium nucleatum*

The genes *atpAGDC* encoding, respectively, the α-, γ-, β- and ε-subunits of the $F_1$-catalytic domain of the ATP synthase from *F. nucleatum* (but lacking the δ-subunit) were cloned into an expression vector with a His$_{10}$-tag at the N-terminus of the ε-subunit and an intervening site for proteolytic cleavage. The purified recombinant enzyme contained the α-, β-, γ- and ε-subunits (figure 1). The specific ATP hydrolytic activity at 37°C of various preparations of the enzyme ranged between 3.5 and 9.4 U mg$^{-1}$ of protein (figure 1; electronic supplementary material, figures S1–S3), similar to specific activities of 4.6–5.8 U mg$^{-1}$ of the intact purified ATP synthase [37]. The apparent $K_m$ value for ATP was 0.12 mM (electronic supplementary material, figure S1). The activity of the enzyme rose with increasing pH from 6.5 to 8.5 where it reached a maximum and declined at higher values (electronic supplementary material, figure S2A). At 4°C, the enzyme was stable over two weeks (electronic supplementary material, figure S2B). At temperatures of 45°C and above, the activity rose substantially, attaining a maximum value of 43.5 U mg$^{-1}$ at 65°C (electronic supplementary material, figure S2C). From 65 to 75°C, the activity declined, consistent with the melting temperature of the enzyme at 72°C (electronic supplementary material, figure S2D). It is known that the c-rings of sodium-dependent ATP synthases are thermostable [37–40], but the exact molecular basis for the thermostability of c-rings and the $F_1$-domain of the *F. nucleatum* enzyme remains unknown, although the general basis of

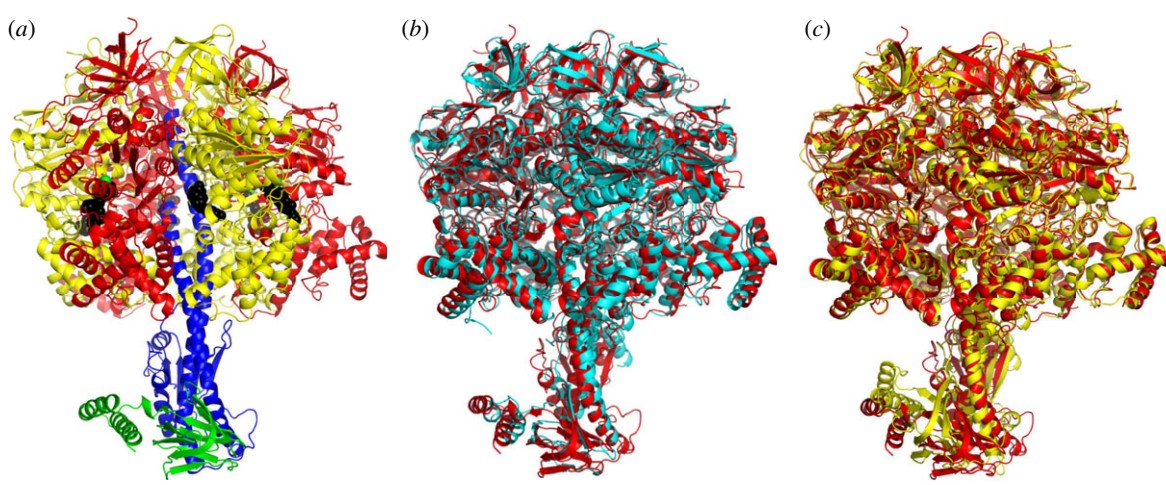

**Figure 2.** Structure of $F_1$-ATPase from *F. nucleatum*. (*a*) Side view of the structure of molecule 1 in ribbon representation with the α-, β-, γ- and ε-subunits in red, yellow, blue and green, and bound nucleotides in a black space-filling representation. The green spheres represent $Mg^{2+}$ ions. (*b*, *c*) Comparison of the structure of the $F_1$-ATPase from *F. nucleatum* (6q45; red) with the structures of $F_1$-ATPases from *M. smegmatis* [51] (6foc; cyan) and *C. thermarum* [52] (5ik2; yellow).

thermostability of proteins is well understood [41,42]. As in $F_1$-ATPases from other species, the ATP hydrolase activity of the enzyme was inhibited by $Mg^{2+}$–ADP [43,44]. Preincubation of the enzyme with 2.5 mM $Mg^{2+}$–ADP inhibited 45% of the ATP hydrolytic activity (figure 1*b*), and similar treatment with 25 mM $Mg^{2+}$–ADP inhibited the enzyme completely. As in other species, the basal hydrolytic activity was stimulated by the addition of lauryldimethylamine oxide [LDAO; 0.05% (w/v)], by a factor of three in this instance (figure 1*c*). As expected, ATP hydrolysis depended on the presence of $Mg^{2+}$ or $Ca^{2+}$ ions, but $Mg^{2+}$ at concentrations in excess of 2.5 mM was partially inhibitory (electronic supplementary material, figure S3). Both $Ca^{2+}$ and $Mg^{2+}$ ions stimulate the hydrolytic activity of the $F_1$-ATPases from *Escherichia coli* [45,46] and from chloroplasts of *Spinacia oleracea* (spinach) [47]. However, the hydrolytic activities of the F-ATPases from *Clostridium paradoxum* [48], from the chemolithotrophic γ-proteobacterium, *Acidithiobacillus ferrooxidans* [49] and from the bacterial thermophile, *Geobacillus stearothermophilus* [43], and bovine $F_1$-ATPases, are lower in the presence of $Ca^{2+}$ than in the presence of $Mg^{2+}$ [50]. By contrast, in the $F_1$-ATPase from *F. nucleatum*, the apparent $K_m$ value of $Ca^{2+}$ was lower, and the apparent $V_{max}$ was higher than with $Mg^{2+}$. However, the intact ATP synthase complex from *F. nucleatum* is not active in the absence of $Mg^{2+}$ or in the presence of $Ca^{2+}$ [37].

## 2.2. Structure of $F_1$-ATPase from *Fusobacterium nucleatum*

Crystals of the $F_1$-ATPase complex from *F. nucleatum* have the unit cell parameters $a = 111.9$ Å, $b = 200.2$ Å, $c = 201.7$ Å, $β = 102.2°$ and belong to the space group $P2_1$ with two $F_1$-complexes in the asymmetric unit (referred to as molecules 1 and 2, respectively; figure 2*a*; electronic supplementary material, figure S4). The structure was solved to 3.6 Å resolution by molecular replacement with the $α_3β_3$-subcomplex, with no ligands or water molecules, taken from the structure of the $F_1$-ATPase from *Caldalkalibacillus thermarum* containing the mutations Asp99Ala and Arg92Ala in the ε-subunit [52]. The statistics for data processing and refinement are summarized in electronic supplementary material, table S1. The quality of the electron density map is indicated in electronic

supplementary material, figure S5, where representative segments and their interpretation are shown. Each of the final models of the two complexes contains three α-subunits ($α_E$, $α_{TP}$ and $α_{DP}$), three β-subunits ($β_E$, $β_{TP}$ and $β_{DP}$), and single copies of the γ- and ε-subunits, and a total of 6420 amino acid residues were resolved. The final model of molecule 1 (figure 2*a*) was better-defined than that of molecule 2 and contained 3218 amino acids distributed between the subunits as follows: $α_E$, 25–500; $α_{TP}$, 27–500; $α_{DP}$, 25–397 and 403–500; $β_E$, 1–460; $β_{TP}$ 1–462; $β_{DP}$, 1–460; γ, 2–282; ε, 1–134. The final model of molecule 2 contains 3202 amino acid residues comprising $α_E$, 27–500; $α_{TP}$, 26–397 and 404–500; $α_{DP}$, 27–400 and 404–495; $β_E$, 2–460; $β_{TP}$ 1–460; $β_{DP}$, 2–460; γ, 2–282; ε, 1–134. The r.m.s.d. value of the superimposed Cα atoms of all subunits of molecule 1 upon molecule 2 was 1.39 Å, and, for the superimposition of Cα atoms of the α- and β-subunits only, the value was 0.46 Å (electronic supplementary material, table S2). The difference between these two values arises from differences in lattice contacts for the two molecules (summarized in electronic supplementary material, figure S4). It results from the different position adopted by the foot of the central stalk relative to the $α_3β_3$-subcomplex in the two molecules, which in turn comes from a lattice contact involving residues 108–113 of the γ-subunit of molecule 2 that is not present in molecule 1. In molecule 2, the foot of the γ-subunit is rotated by about 12° in a clockwise direction, as viewed from above the 'crown' towards the membrane domain of the intact ATP synthase.

## 2.3. Catalytic $α_3β_3$-domain

As in other structures of $F_1$-ATPases, each α-subunit and each β-subunit have three domains: an N-terminal domain with six β-strands, a central nucleotide-binding domain and a C-terminal domain consisting of a bundle of α-helices (seven in the α-subunits and four in the β-subunits). The six N-terminal domains associate together to form the 'crown' of the complex. The $α_3β_3$-domain and the entire $F_1$-ATPase from *F. nucleatum* were compared with the equivalent structures from *Mycobacterium smegmatis* (6foc) [51] and *C. thermarum* (5ik2) [52] (figure 2*b*,*c*; electronic supplementary material, table S3). Similar comparisons were made with the $α_3β_3$-domains and $F_1$-ATPases from chloroplasts in spinach

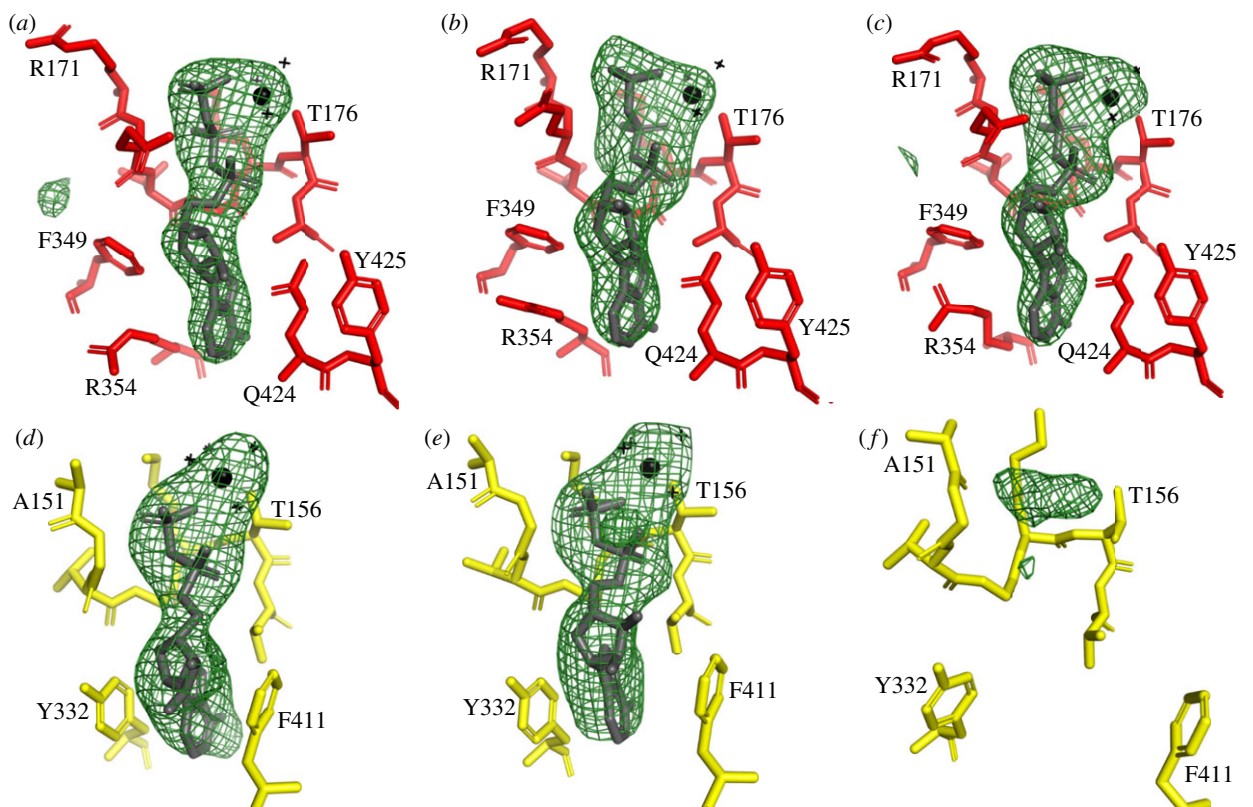

**Figure 3.** Occupancy of nucleotide-binding sites in the α- and β-subunits of the $F_1$-ATPase from *F. nucleatum*. An $F_o - F_c$ difference density map for the complex was calculated with the nucleotides, $Mg^{2+}$ and water molecules at zero occupancy. The green mesh represents the difference density in the six nucleotide-binding sites contoured to 3.0 σ. In (*a–c*), the $α_{DP}$-, $α_{TP}$- and $α_E$-subunits; in (*d–f*), the $β_{DP}$-, $β_{TP}$- and $β_E$-subunits from molecule 1. In (*a–c*), the sites are occupied by an ATP molecule and an accompanying $Mg^{2+}$ (black sphere) with three water ligands (black crosses); the fourth, fifth and sixth ligands are provided by O2B and O2G of the ATP and the hydroxyl of αThr-176. In (*d,e*), the sites are occupied by an ADP molecule and an accompanying $Mg^{2+}$ (black sphere) with four water ligands (black crosses); the fifth and sixth ligands are provided by O2B of the ADP and the hydroxyl of βThr-156. In (*f*), the difference density in the vicinity of the P-loop cannot be interpreted with confidence, but it probably can be accounted for by an ADP molecule (without $Mg^{2+}$) at low occupancy or citrate.

(6fkf) [53], *Paracoccus denitrificans* (5dn6) [54], four bovine structures (4yxw, 2jdi, 1e79, 4asu) [10,12,15,23] and the $α_3β_3$-domains from *G. stearothermophilus* (4xd7) [55] and *E. coli* (3oaa) [56] and with entire $F_1$-domains from the same species (electronic supplementary material, table S3). The most similar $α_3β_3$-domains were those from *M. smegmatis* (6foc) [51] and *C. thermarum* (5ik2) [52], and the structures of their $F_1$-domains were the most closely related also. The least similar $α_3β_3$-domains were those from *G. stearothermophilus* (4xd7) [55] and *E. coli* (3oaa) [56]. Likewise, the least related $F_1$-domains were also from *G. stearothermophilus* (4xd7) [55] and *E. coli* (3oaa) [56]. The high r.m.s.d. values for the $F_1$-domains from *G. stearothermophilus* (4xd7) [55] and *E. coli* (3oaa) [56] arise because their ε-subunits are in the 'up' conformation, where the two C-terminal α-helices lie along-side the α-helical coiled-coil in the γ-subunit, resulting in the $α_{DP}–β_{DP}$ interface being displaced outwards (see below).

The nucleotide-binding sites in the three α-subunits have additional electron density that is compatible with each of them being occupied by an ATP molecule and an accompanying magnesium ion (figure 3*a–c*). Similarly, additional density in the nucleotide-binding sites of the $β_{TP}$- and $β_{DP}$-subunits provides strong evidence for the presence in each site of an ADP molecule with a magnesium ion (figure 3*d,e*). There is also some density in the nucleotide-binding site of the $β_E$-subunit (figure 3*f*), which is increased slightly in molecule 2 relative to molecule 1 (electronic supplementary material, figure S6), although the amino acid side chains that form the

nucleotide-binding sites in the two molecules are essentially identical positions. Possible interpretations of this density are either that it is an ADP molecule at very low occupancy, or a citrate molecule, or a mixture of both (electronic supplementary material, figure S6). Citrate was present in the crystallization buffer, and it fits the density in molecule 1 better than in molecule 2 (electronic supplementary material, figure S6E,F). It has been found to be bound to the P-loop of RecA from *M. smegmatis* [57]. Although the crystallization buffer contained 500 μM ADP, similar to the conditions used with the $F_1$-ATPase from *C. thermarum* (5ik2) [52] and *M. smegmatis* (6foc) [51], no ADP was added to the buffer for harvesting the crystals of the *F. nucleatum* $F_1$-ATPase, and its absence probably accounts for the low occupancy in this site compared to the *C. thermarum* and *M. smegmatis* enzymes. The interpretation of the current data is not certain, and therefore neither a nucleotide nor citrate has been included in this site in the model. There is no evidence for the binding of either a magnesium ion or phosphate in the $β_E$-subunit of $F_1$-ATPase in *F. nucleatum*.

## 2.4. γ-subunit

In the structure of the $F_1$-ATPase from *F. nucleatum*, the γ-subunit was resolved completely. As in other $F_1$-ATPases, it has five α-helices, αH1–αH5. Helices αH1 and αH5 make an antiparallel, α-helical coiled-coil that occupies the central axis of the $α_3β_3$-domain, and αH2–αH4 are part of a Rossmann

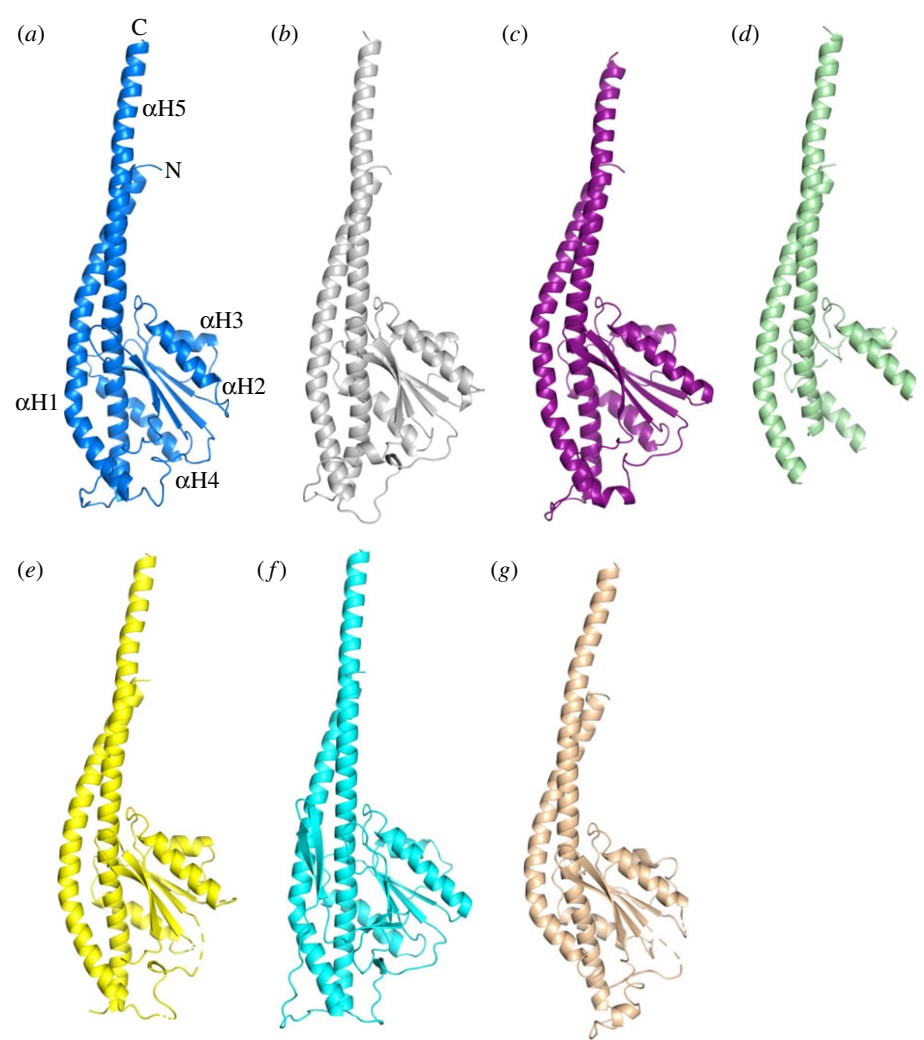

**Figure 4.** Comparison of the structure of the γ-subunit of the F-ATPase from *F. nucleatum* with those of orthologues. (*a*) *F. nucleatum* (6q45; molecule 1) with the five α-helices numbered 1–5 from N- to C-terminus; (*b*) *C. thermarum* (5ik2) [52]; (*c*) *E. coli* (3oaa) [56]; (*d*) *M. smegmatis* (6foc) [51]; only the α-helices were resolved; (*e*) *P. denitrificans* (5dn6) [54]; (*f*) spinach chloroplasts (6fkf) [53]; and (*g*) bovine mitochondria (1e79) [15].

fold with five β-strands with the three α-helices between strands 1 and 2, 2 and 3, and 3 and 4. The lower part of the coiled-coil interacts with the N-terminal domain of the ε-subunit. Superimposition of the *F. nucleatum* γ-subunit on orthologues showed that it is most similar to bacterial γ-subunits from *C. thermarum* (5ik2) [52], *E. coli* (3oaa) [56], *P. denitrificans* (5dn6) [54] and also to the fragmentary structure of the γ-subunit from *M. smegmatis* (6foc) [51], and to a lesser extent to the γ-subunit in spinach chloroplasts (6fkf) [53] where αH1 is straighter, and the subunit has the additional β-hairpin involved its redox-linked regulatory mechanism (see below). The overall fold of these bacterial γ-subunits (figure 4) is also similar to that of the γ-subunits from the enzymes from bovine (1e79) [15] and yeast (2hld) [58] mitochondria, although in the bacterial subunits αH1 extends further in a C-terminal direction and is less curved.

The rotation of the γ-subunit drives the synthesis of ATP in the $F_1$-domains of F-ATP synthases with energy provided by the pmf (or smf), and each 360° rotation in three 120° steps generates three ATP molecules, one from each of the three catalytic sites of the enzyme [1,2]. During the hydrolysis of ATP, the energy provided by the ATP molecule drives rotation in the opposite sense. The hydrolytic 360° cycle also has three 120° steps [59,60], and the intervening pauses are known as the 'catalytic dwells', where the enzyme is poised to carry out, or

is carrying out, ATP hydrolysis. At lower concentrations of ATP, a second pause, known as the 'ATP-binding dwell' [61,62], where the enzyme awaits the binding of the substrate, occurs 40° after the catalytic dwell, and in the mitochondrial enzyme, but not in bacterial enzymes, a third pause, the 'phosphate release dwell', has been observed by stopping rotation with the phosphate analogue, thiophosphate, 25° before the catalytic dwell [63].

In the wide range of high-resolution structures of $F_1$-domains that have been resolved, the majority being structures of the bovine enzyme inhibited in a variety of ways, the 'foot' of the γ-subunit consisting of the Rossmann-fold domain and the associated antiparallel α-helical coiled-coil of αH1 and αH5, has rotated to a range of positions. By contrast, the 'upper' part of the γ-subunit, consisting of the N-terminal region of αH1, occupies the same position because of the intervention of a 'catch loop' provided by the adjacent $β_E$-subunit [64]. This 'catch loop' holds the γ-subunit and allows torsional energy to be stored somewhere below the catch. Once a critical point is reached, the stored energy is released in a quantum to generate the rotational step or sub-step. The role of the 'catch loop' is illustrated in electronic supplementary material, figure S7, where a selection of mitochondrial and bacterial structures, including the current one from *F. nucleatum*, have been superimposed. The rotational positions of the 'foot' domains in the various structures are summarized in electronic

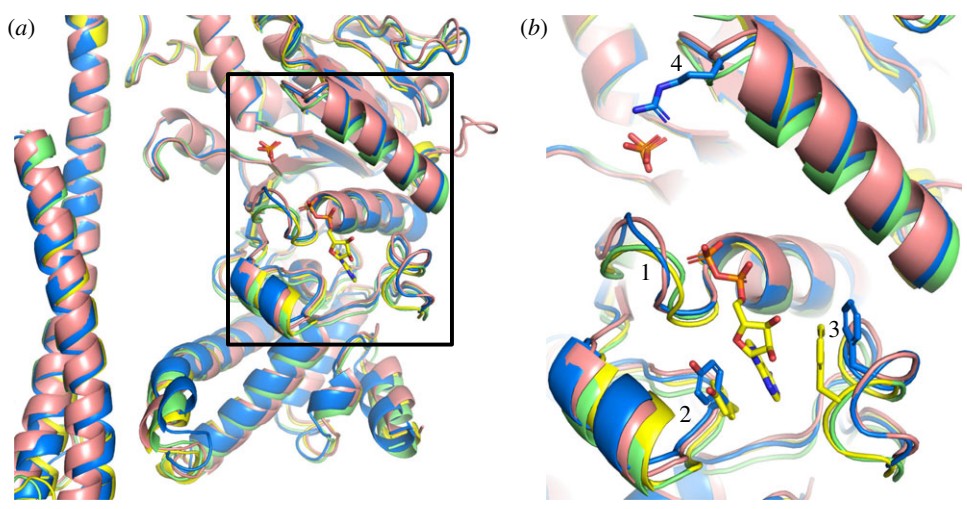

**Figure 5.** Comparison of the nucleotide-binding sites in $\beta_E$-subunits in various bacterial $F_1$-ATPases. (*a*) Cartoon representation of part of the $\alpha$-helical coiled-coil of the $\gamma$-subunits and adjacent nucleotide-binding domains and C-terminal $\alpha$-helical domains of $\beta_E$-subunits based on the superimposition of $F_1$-ATPases via their crown domains; *F. nucleatum* (6q45; blue); *P. denitrificans* [54] (5dn6; pink); *M. smegmatis* [51] (6foc; green); *C. thermarum* [52] (5ik2; yellow). An ADP molecule bound to the $\beta_E$-subunit from *C. thermarum*, and phosphate ions bound to the $\beta_E$-subunits from *C. thermarum* and *M. smegmatis* are shown in the stick representation. (*b*) Magnified version of the region in the box in (*a*); regions 1, residues 151–154 towards the N-terminal end of the P-loop (residues 149–156) in *F. nucleatum*; regions 2 and 3 contain aromatic residues, Tyr-332 and Phe-411 in *F. nucleatum* (shown in the blue stick representation), that form a pocket where the adenine ring of ADP binds; the equivalent residues in *C. thermarum* (Tyr-334 and Phe-413) are shown in yellow; region 4, loop containing an arginine residue (Arg-182 from *F. nucleatum* shown in the blue stick representation) involved in binding phosphate ions in *M. smegmatis* and *C. thermarum*, but not evidently in *F. nucleatum* and *P. denitrificans*.

supplementary material, table S4 and figure S8, with the $\gamma$-subunit in the original 'ground-state' structure assigned arbitrarily as having a rotation of $0°$ [10] (see Material and Methods for the measurement of rotation; electronic supplementary material, table S4). They include structures that can be related plausibly to rotational positions observed in 'single-molecule' experiments with human $F_1$-ATPase [63]. In these rotational experiments, the phosphate release dwell is defined by the position adopted by the central stalk when rotation is inhibited by the phosphate analogue, thiophosphate, and in similar experiments, the $F_1$-ATPase inhibitor protein $IF_1$ stopped rotation at the catalytic dwell. Therefore, the structure of bovine $F_1$-ATPase inhibited with thiophosphate (4yxw) [9] provides a structural representation of the phosphate release dwell, and structures of bovine $F_1$-ATPase inhibited by the monomeric form of $IF_1$ consisting of residues 1–60 (4tt3, 4tsf, 2v7q) [9,16] describe the catalytic dwell. Moreover, in 'ground-state' structures, for example (1bmf, 2jdi) [7,12], the enzyme has been arrested at approximately the same rotary position as in the thiophosphate-inhibited state [10], and therefore, it can also be interpreted as representing the phosphate release dwell. Likewise, in the structure of bovine $F_1$-ATPase crystallized in the presence of phosphonate (4asu), rotation has been arrested at the same rotary position at approximately $30°$ as in the $IF_1$-inhibited enzyme, and so it can be ascribed as representing the catalytic dwell [23]. Neither the structural data nor the 'single-molecule' rotary experiments [63] support alternative proposals based on simulations [65] that the structure of bovine $F_1$-ATPase crystallized in the presence of phosphonate (4asu) [23] represents the ATP-binding dwell and that the 'ground-state' structures (e.g. 1bmf, 2jdi) [7,12] represent the catalytic dwell [65]. It is possible, but not certain, that the ATP-binding dwell is represented by the structure of bovine $F_1$-ATPase inhibited by ADP and aluminium fluoride (1h8e) [19], where the $\gamma$-subunit has rotated through $105°$ (electronic supplementary material, figure S8 and table S4).

Based on the rotations of their $\gamma$-subunits, the structures of those bacterial $F_1$-ATPases where their hydrolytic activities seem to be regulated by the failure to release one or more of the products of hydrolysis, namely *M. smegmatis* ($10.5°$ rotation) [51] and *C. thermarum* ($11.7°$ and $13.2°$ for molecules 1 and 2, respectively) [52], lie at the position of the phosphate release dwell in the mammalian enzyme. *Fusobacterium nucleatum* ($19.7°$ and $19.1°$ for molecules 1 and 2, respectively) falls between the position of the phosphate release and catalytic dwells in the mammalian enzyme (electronic supplementary material, table S4). The hydrolytic activity of the *P. denitrificans* enzyme is inhibited by the $\zeta$-subunit, an orthologue in its inhibitory region of the inhibitory region of bovine $IF_1$, and therefore it is likely that the *P. denitrificans* structure with a $\gamma$-subunit rotation of $27°$ represents the catalytic dwell of the enzyme. Single-molecule experiments conducted with $F_1$-ATPases from *G. stearothermophilus* and *E. coli* show that phosphate is released at the end of the catalytic dwell [66] and that ADP is released $25°$ before the catalytic dwell [67]. These positions approximate to the rotary positions in the human enzyme [63], but in the reverse order.

A comparison of the structures of the nucleotide-binding sites of the $\beta_E$-subunits of $F_1$-ATPases from *F. nucleatum* with those in the $F_1$-ATPases from *P. denitrificans* (5dn6) [54], *M. smegmatis* (6foc) [51] and *C. thermarum* (5ik2) [52] (figure 5) illustrates that in this region, the *F. nucleatum* subunit is most similar to the *P. denitrificans* subunit, and that the *C. thermarum* and *M. smegmatis* proteins provide a second similar pair, that has a somewhat different conformation to the *F. nucleatum* and *P. denitrificans* subunits. This progression from the least open to the most open $\beta_E$-subunit corresponds with the order of the extents of rotation of the $\gamma$-subunit (electronic supplementary material, table S4). The differences between the two pairs are most marked in regions 1–4 in figure 5*b*. Region 1 (*F. nucleatum* residues 151–154) is part of

royalsocietypublishing.org/journal/rsob    Open Biol. 9: 190066

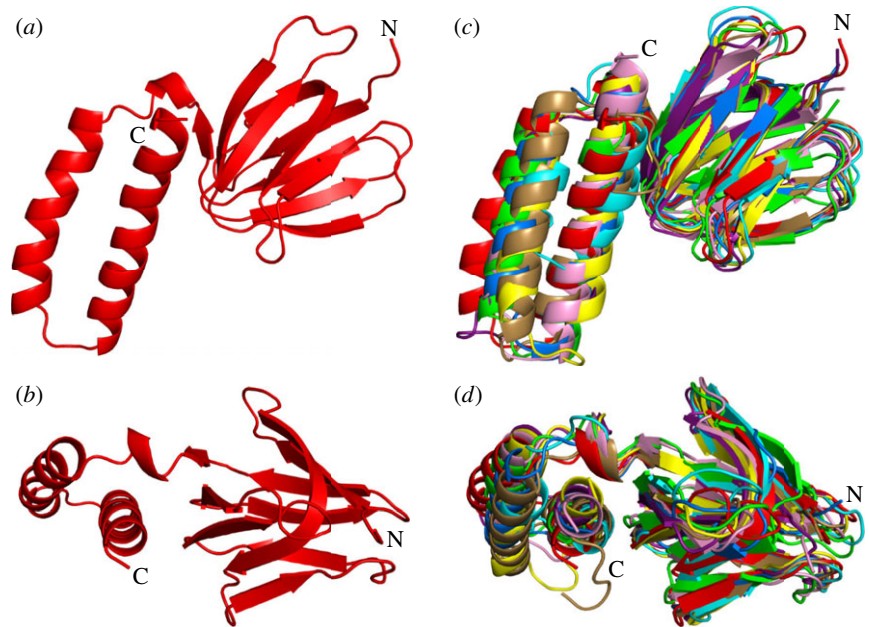

**Figure 6.** Comparison of the structure of the ε-subunit from *F. nucleatum* with those of orthologues. (*a,b*) The *F. nucleatum* ε-subunit (6q45, red) viewed from beneath the α₃β₃-domain along the axis of the central stalk and rotated by 90°, respectively; (*c,d*) the same views as in (*a*) and (*b*) with the structures of ε-subunits from the following species superimposed; *M. smegmatis* [51] (6foc; cyan); *C. thermarum* [52] (5ik2; yellow); *E. coli* [68] (1aqt; pink); *G. stearothermophilus* [69] (2e5y; purple); *S. oleracea* [53] (6fkf; marine blue); and *T. elongatus* [70] (5zwl; wheat); and with the δ-subunit from bovine mitochondrial F₁-ATPase [15] (1e79;green).

the P-loop, and in both *F. nucleatum* and *P. denitrificans*, it is displaced away from the γ-subunit relative to *C. thermarum* and *M. smegmatis*. Region 2 in *F. nucleatum* and *P. denitrificans* is displaced towards the γ-subunit relative to the *C. thermarum* and *M. smegmatis* subunits, and contains Tyr-332 (*F. nucleatum* numbering). However, this tyrosine residue (replaced by Phe-343 in *M. smegmatis*) contributes to one side of the adenine-binding pocket and is in approximately the same position in the four structures. By contrast, in region 3 of the *F. nucleatum* βE-subunit, Phe-411 (and the equivalent Phe-420 in *P. denitrificans*), which contributes to the opposite side of the adenine-binding pocket, is displaced away from the pocket by about 4 Å outwards relative to the equivalent residue, Phe-413, in *C. thermarum*, and therefore, the nucleotide would be expected to bind less strongly in *F. nucleatum* than in *C. thermarum*, as the structure of the *F. nucleatum* F₁-ATPase suggests. The *P. denitrificans* enzyme was crystallized in the presence of ATP only, and ATP has never been observed bound to a βE-subunit in any structure of F₁-ATPase. In region 4, the βE-subunit of *F. nucleatum* is more similar to the *C. thermarum* protein than to the *P. denitrificans* and *M. smegmatis* proteins. This region forms a loop leading into the α-helix in the top-right of figure 5b. In this loop is found residue Arg-182 (*F. nucleatum* numbering). In *C. thermarum* and *M. smegmatis*, this residue helps to coordinate the bound phosphate, and in *F. nucleatum*, the side chain is in a similar position, and yet no phosphate is evidently bound in this site. Currently, there is no clear explanation for why phosphate is not bound also in the *F. nucleatum* F₁-ATPase.

## 2.5. ε-subunit

The N-terminal domain of the ε-subunit is folded into a 10-stranded β-sandwich, and the C-terminal domain consists of a hairpin of two α-helices, lying alongside the β-sandwich (figure 6). This conformation of two α-helices is known as the

'down' position. Superimposition of the structures of the ε-subunit from *F. nucleatum* via their N-terminal domains on those of orthologues demonstrates that the α-helices are in a similar position to those in *E. coli* (1aqt, 1bsn) [68,71,72], *G. stearothermophilus* (2e5y) [69] and *C. thermarum* (5ik2 and 5hkk) [52] (see electronic supplementary material, table S3). Similar to the structures of ε-subunits determined in the context of the intact F₁-ATPases from *E. coli* (3oaa) [56] and *M. smegmatis* (6foc) [51], and in the structures of the intact ATP synthase from *E. coli* (5t4o) [73], no ATP molecule was bound to the ε-subunit in the F₁-ATPase from *F. nucleatum*, and none was bound to the isolated ε-subunit or to the ε-subunit in the γε-subcomplex from *Thermosynechococcus elongatus* (5zwl) [70]. However, an ATP molecule with an accompanying Mg²⁺ ion has been found bound to the ε-subunit in the F₁-ATPase from *C. thermarum* (5hkk) [52] and to the isolated ε-subunit in *G. stearothermophilus* (2e5y) [69]. In the ε-subunits in *C. thermarum* (5hkk) [52], *E. coli* (3oaa, 1aqt) [56,68] and *G. stearothermophilus* (4xd7, 2e5y) [55,69], four conserved amino acids are involved in binding ATP [74]. They are Ile-88, Asp-89, Arg-92 and Ala-93 (*C. thermarum* numbering; see electronic supplementary material, figure S9). In *F. nucleatum*, the first two residues are conserved, but the arginine and alanine residues are replaced by serine and glutamic acid, respectively, and hence, the *F. nucleatum* ε-subunit lacks essential features for binding an ATP molecule at this site (electronic supplementary material, figure S9). As described below, the ε-subunit has been studied extensively in the context of regulating the hydrolytic activity of bacterial ATP synthases.

## 2.6. Regulation of bacterial ATP synthases

Eubacteria have evolved a variety of mechanisms for regulating the hydrolytic activity of their ATP synthases. In α-proteobacteria, exemplified by *P. denitrificans*, ATP hydrolysis appears to be inhibited by a protein called the ζ-subunit [54,75],

where the N-terminal inhibitory region binds to a catalytic interface under hydrolytic conditions in a closely related manner to the inhibitory action of the orthologous mitochondrial regulatory protein IF₁ on the mitochondrial ATP synthase [9,16,76]. Cyanobacterial ATP synthases are regulated by a mechanism that appears to be similar, but not identical, to the way that ATP synthases in the chloroplasts of green plants and algae are regulated. In the absence of light, when the pmf is low, ADP–Mg$^{2+}$ remains bound to one of the three catalytic sites of the chloroplast enzyme forming an inactive ADP-inhibited state of the enzyme [77,78]. This inhibited state is reinforced by the formation of an intramolecular disulfide bond in the γ-subunit of the enzyme, which is thought to stabilize a β-hairpin structure formed by a unique additional sequence (residues 198–233) in the γ-subunit. This β-hairpin wedges between the β-subunit and the central stalk, and may suppress futile ATP hydrolysis by preventing the rotation of the γ-subunit [53]. When light is restored, the pmf increases and reduction of the disulfide bond by thioredoxin unlocks the ATP synthetic activity of the enzyme. In cyanobacterial ATP synthases, the γ-subunits also contain a related insertion [79] that appears to inhibit ATP hydrolysis [70], but it lacks the segment containing the two cysteine residues, and so it cannot be regulated by a similar redox mechanism [80].

The role of the ε-subunit in the regulation of bacterial and chloroplast ATP synthases is an area of active study. The known structures of all bacterial [52,55,68,69,71–73,81] and chloroplast [53,70] ε-subunits, and the orthologous δ-subunit [15,58] in mitochondria, consist of an N-terminal domain folded into a 10-stranded β-sandwich, and a C-terminal domain folded into an α-helical hairpin. In the intact enzyme, the β-sandwich domain is involved in binding the ε-subunit to both the γ-subunit in the central stalk of the F₁-domain and the c-ring in the membrane domain of the enzyme. In the various structures of ATP synthases and F₁-ATPases, the α-helical C-terminal region has been observed in one of two different conformations. In most structures, the two α-helices are associated closely with the β-sandwich, in the 'down' conformation, with an ATP molecule bound between the two domains, in the case of *C. thermarum* (5hkk) [52] and *G. stearothermophilus* (2e5y) [69]. In the intact ATP synthases from *E. coli* (5t4o) [73] and *G. stearothermophilus* (6n2y) [82] and in the F₁-domain from *E. coli* (3ooa) [56] and *G. stearothermophilus* (2e5y) [55], the two α-helices assume a different 'up' conformation, where they penetrate into the α₃β₃-catalytic domain along the axis of the coiled-coil of the N- and C-terminal α-helices of the γ-subunit [55,56,73,82]. In this conformation, the N-terminal domain has no bound ATP molecule. Therefore, it has been proposed that in the enzyme from *G. stearothermophilus* [55] but not in the *E. coli* enzyme, that a 'down'-'up' switch might provide a physiological mechanism that operates when the pmf and the concentration of ATP are low [69,74,83,84]. Under these conditions, the ATP molecule would leave the ε-subunit, allowing the two α-helices to dissociate from the β-domain and form the inhibitory 'up' conformation. In the thermophilic cyanobacterium, *T. elongatus*, where no nucleotide was observed bound to the isolated subunit (2ro6) [85], and in another structure of the γε subcomplex where no nucleotide was present during the crystallization of the subcomplex (5zwl) [70], the ε-subunit was down in both instances. In this organism, it has been proposed that the ATPase is regulated by the γ-subunit, in a similar fashion to the regulation of ATP hydrolysis in the chloroplast enzyme, but without the regulation via the oxidation and reduction of

a disulfide linkage that occurs in the chloroplast enzyme [70]. However, the ATP synthases from *C. thermarum* [52] and *M. smegmatis* [51] appear not to conform to this mechanism of regulation. They can both synthesize ATP under appropriate conditions, but they hydrolyse ATP very poorly. The structures of their F₁-catalytic domains are very similar to each other and also to the F₁-domain of the ATP synthase from *F. nucleatum*, but despite being inhibited in ATP hydrolysis, the ε-subunit of the *C. thermarum* enzyme is in the 'down' position with an ATP molecule and a magnesium ion bound to it. Moreover, the subunit remained in the 'down' position, and the enzyme remained inhibited when the capacity of the ε-subunit to bind an ATP molecule was removed by mutation [52]. In the mycobacterial enzyme, the α-helical hairpin of its ε-subunit is truncated and incapable of binding an ATP molecule, and it also is in the 'down' position [51]. However, in the structures of the F₁-catalytic domains from *C. thermarum* and *M. smegmatis*, a phosphate (possibly a sulfate in *M. smegmatis*) is bound to the most open of the three catalytic sites, suggesting that the hydrolytic activity of this enzyme may be inhibited by the failure to release one of the products of hydrolysis. In the same catalytic site, ADP is bound also in *C. thermarum* and is possibly present at low occupancy in *M. smegmatis*. In *F. nucleatum*, the ε-subunit is 'down' with no bound ATP, but, in contrast to the F₁-ATPases from *C. thermarum* and *M. smegmatis*, it is intrinsically active in ATP hydrolysis, and that activity can be stimulated by LDAO. This behaviour is reminiscent of the behaviour of F₁-ATPases from mitochondria, which are similarly intrinsically active and their activity is also stimulated by LDAO. One explanation of the stimulatory effect of LDAO is that it releases inhibitory Mg$^{2+}$–ADP from the catalytic sites [43] and it has a similar effect on the α₃β₃γ-subcomplex from *G. stearothermophilus* [86,87]. However, in the F₁-ATPase from *C. thermarum*, this appears not to be the complete explanation as the activity of this enzyme, in addition to being stimulated by LDAO, is also partially activated by the removal of the C-terminal domain of the ε-subunit and could then be activated to its fullest extent by LDAO [88]. In the *E. coli* F₁-ATPase, where the ε-subunit is permanently 'up' [56,73,89], LDAO has an additional effect as it influences interactions between the catalytic β-subunit and the ε-subunit. Thus, currently, the most plausible interpretation of the structure of the enzyme from *F. nucleatum* is that it represents the state of the enzyme that is partially inhibited by Mg$^{2+}$–ADP, similar to the bovine F₁-ATPase, and this partial inhibition can be relieved by LDAO. The exact molecular role of LDAO in activating these various F₁-ATPases remains obscure. One possibility is that it loosens the structure of the nucleotide-binding domain so that the nucleotide is bound less tightly, and a similar explanation can be advanced to explain the activation of the *F. nucleatum* F₁-ATPase by increased temperatures up to 65°C. Structures of LDAO-activated F₁-ATPases might help to resolve this issue.

# 3. Material and methods

## 3.1. Bacterial strains

*Escherichia coli* DH10B [90] and MC1061, used in cloning experiments, were grown in LB medium (10 g l$^{-1}$ tryptone, 5 g l$^{-1}$ yeast and 5 g l$^{-1}$ NaCl). The overexpression strain *E. coli* DK8 (*Δunc*) [91] was grown in medium containing 2× YT (16 g l$^{-1}$ tryptone, 10 g l$^{-1}$ yeast extract and 5 g l$^{-1}$ NaCl)

royalsocietypublishing.org/journal/rsob   *Open Biol.* **9**: 190066

royalsocietypublishing.org/journal/rsob    Open Biol. 9: 190066

plus 0.2% [w/v] glucose to compensate for the absence of a functional ATP synthase.

## 3.2. Construction of expression plasmids

The genes *atpAGDC* from *F. nucleatum* encoding the α-, γ-, β- and ε-subunits, respectively, of ATP synthase were amplified by a polymerase chain reaction from genomic DNA with the primers FusoF1for (5′-TTTTCCATGGATGAATATTAGAC-CAGAAGAAG-3′) and FusoF1rev (5′-TTTTGGATCCTTAA TTATTCTTAGCATCTATTTTTG-3′). The product was cloned into the expression vector pTrc99a (Amersham Biosciences). The translational initiation codons of the α- and β-subunits were changed to ATG, generating the expression construct pJP2. To facilitate the purification of enzyme, the sequence encoding either a $His_{10}$-tag with a following cleavage site for the protease from tobacco etch virus (TEV) (pJP3) or a $His_{10}$ followed by a 6-residue (Ser-Gly-Gly-Gly-GlyGly) linker, an intervening TEV protease cleavage site and another 6-residue (Ser-Gly-Gly-Gly-GlyGly) linker (pJP5) were introduced at the 5′-end of *atpC* encoding the ε-subunit. For pJP3, the primers FusoHis_F1for (5′-TTTTT GAATTCCATCTGCTGTTGGATATCAACC-3′) and Fuso-His_F2rev (5′-AAGATTCTCATGGTGATGGTGATGGTGAT GGTGATGGTGCATATTCCCTCCTTATTTTGCTAAATC-3′) were used for the first fragment and FusoHis_F3for (5′-CAT-CACCATCACCATCACCATGAGAATCTTTATTTTCAGGGC CCTAGTTTTGATGTAAGTGTTGTAACAC-3′) and Fuso-F1rev for the second fragment. For pJP5, the primers FusoHis_F1for and FusoHisLinker_F2rev (5′-**ACCTGAGCC CTGAAAATAAAGATTCTCACC**GCCACCGCCACCTGAA TGGTGATGGTGATGGTGATGGTGATGGTGCATATTCCCT CCTTATTTTGCTAAATC-3′) for generating the first fragment and FusoHisLinker_F3for (5′-**CACCATCACCATCACCATT CAGGTGGCGGT**GGCGGTGAGAATCTTTATTTTCAGGG CTCAGGTGGCGGTGGCGGTCCTAGTTTTGATGTAAGTG TTGTAACAC-3′) and FusoF1rev for amplifying the second fragment. In both cases, the two fragments overlapped by 30 nucleotides and were joined by overlap extension with the external primers FusoHis_F1for and FusoF1rev. The resulting fragments were cloned into EcoRI and BamHI sites in pJP2 producing the expression vectors pJP3 and pJP5. The sequences of all four genes were verified by DNA sequence analysis. A protein expressed from pJP3 was used in all assays. However, it was found that the TEV protease was unable to cleave the $His_{10}$-tag. The protein used in the crystallization trials was expressed from pJP5 where the $His_{10}$-tag was able to be removed by the TEV protease.

## 3.3. Expression and purification of $F_1$-ATPase from *Fusobacterium nucleatum*

Expression plasmids pJP3 and pJP5 were transformed into *E. coli* DK8 (*Δunc*), together with the helper plasmid pRARE (Addgene). The cells were grown at 37°C to an optical density of 0.4–0.8 at 600 nm in 2× YT medium plus ampicillin (100 μg ml$^{-1}$), chloramphenicol (34 μg ml$^{-1}$) and 0.2% [w/v] glucose. Expression from the *trc*-promoter was induced with 1 mM isopropyl β-D-1-thiogalactopyranoside, and the culture was incubated for 3–4 h at 37°C and then for 16 h at 30°C. The cells were harvested and washed with buffer

(50 mM Tris–HCl pH 8.0 and 2 mM $MgCl_2$) and either used immediately or stored at −20°C. The yield of wet cells was 2 g l$^{-1}$. Cells (approx. 7–10 g) were resuspended in the same buffer plus cOmplete EDTA-free protease inhibitor tablets (Roche) and DNase I (Roche), and disrupted by two passages through a Constant Systems cell disrupter at 31 kpsi. Cell debris was removed by centrifugation (10 000 × *g*, 15 min, 4°C), and the supernatant was centrifuged again (131 500 × *g* for 45 min at 4°C). To the resulting supernatant, 100 mM NaCl and 25 mM imidazole were added, and this solution was loaded at a flow rate of 2 ml min$^{-1}$ onto a HisTrap HP nickel affinity column (5 ml; GE Healthcare). The column was washed with buffer A consisting of 20 mM Tris–HCl, pH 8.0, 10% [w/v] glycerol, 2 mM $MgCl_2$, 100 mM NaCl, 25 mM imidazole, and 0.1 mM phenylmethylsulfonyl fluoride. The $F_1$-ATPase was eluted with a linear gradient of 100 ml of buffer A and buffer A containing 500 mM imidazole. For use in enzymic or biophysical experiments, the enzyme was pooled and concentrated by ultrafiltration with a 100 kDa cut-off membrane, and then passed through a Superose 6 10/300 size exclusion column (GE Healthcare), equilibrated in buffer consisting of 20 mM Tris–HCl, pH 8.0, 10% [w/v] glycerol, 2 mM $MgCl_2$, and 100 mM NaCl at a flow rate of 0.5 ml min$^{-1}$. Fractions containing $F_1$-ATPase were pooled. For use in crystallization experiments, fractions containing $F_1$-ATPase from the HisTrap column were pooled and the $His_{10}$-tag was cleaved off with the TEV protease for 18 h at 23°C in buffer containing 20 mM Tris–HCl, pH 8.0, 20% [w/v] glycerol, 2 mM $MgCl_2$, 100 mM NaCl, and 1 mM tris(2-carboxyethyl)-phosphine). The sample was concentrated to 2–3 ml by centrifugal ultrafiltration (100 kDa molecular mass cut-off) and then re-loaded onto the HisTrap HP column at a flow rate of 1 ml min$^{-1}$. The $F_1$-ATPase eluted in the excluded volume of the column. Fractions containing the enzyme were pooled, concentrated and applied to a Superose 6 10/300 size exclusion column (GE Healthcare) equilibrated in buffer containing 20 mM Tris–HCl, pH 8.0, 10% [w/v] glycerol, 2 mM $MgCl_2$, 100 mM NaCl, and 1 mM ADP.

## 3.4. Biochemical methods

The $F_1$-ATPase from *F. nucleatum* was analysed by SDS–PAGE on 4–12% NuPAGE Bis–Tris Mini gels (Life Technologies). Proteins were detected with Coomassie G-250 dye. The bands from the stained gel were excised, and the identities of the proteins were verified by mass-mapping of tryptic peptides in a MALDI-TOF mass spectrometer. Protein concentrations were measured with the DC protein assay kit (Bio-Rad) with bovine serum albumin as a standard. ATP hydrolysis was measured by an ATP-regenerating assay, at 37°C unless otherwise stated, where one unit of activity is equal to 1 μmol of ADP produced per minute [92] or by the colorimetric assay of inorganic phosphate where one unit of activity is equal to 1 μmol of phosphate released per minute [93]. The influence of pH on the activity of the enzyme was examined in a three-buffer mixture composed of 50 mM each of MES–MOPS–Tris–HCl [94]. The inhibitory effect of increasing concentrations of $Mg^{2+}$–ADP at a constant ratio of $Mg^{2+}$ : ADP 2 : 1 (w : w) on ATP hydrolysis was examined. The enzyme was pre-incubated with the $Mg^{2+}$–ADP mixture for 10 min and then ATP hydrolysis was initiated by the addition of ATP.

## 3.5. Thermal stability

The melting temperature of $F_1$-ATPase was determined in a LightCycler 480 (Roche) in a reaction mixture (20 μl) containing 5 μM $F_1$-ATPase in buffer consisting of 20 mM Tris–HCl, pH 8.0, 10% [w/v] glycerol, 2 mM $MgCl_2$, and 100 mM NaCl, 5× SYPRO Orange Dye and 100 mM Tris–HCl, pH 8.0, at 20°C. The assay [95,96] was optimized with enzyme concentrations of 0.5–5 μM and 5–20× SYPRO Orange Dye. The optimal conditions were 5 μM $F_1$-ATPase and 5× SYPRO Orange Dye. Samples were equilibrated at 20°C for 5 min, and then the temperature was increased by 1°C min$^{-1}$ to 95°C. Fluorescence was measured at intervals of 0.588°C, and the melting point of $F_1$-ATPase was calculated with LightCycler 480 software v.1.5.1.62.

## 3.6. Crystallization of $F_1$-ATPase from *Fusobacterium nucleatum*

The enzyme was concentrated by ultrafiltration to 2–2.5 mg ml$^{-1}$ and centrifuged (16 000 × *g*, 5 min) at 4°C. It was crystallized at 18°C by vapour diffusion in hanging drops in 24-well plates. The drops consisted of 1 μl of protein solution, 0.8 μl of precipitant buffer [100 mM sodium citrate, pH 6.0, 100 mM magnesium acetate and 15.5% [w/v] polyethylene glycol 5000 monomethyl ether] and 0.2 μl of low melting-point agarose (Hampton Research) (final concentration 0.2% [w/v]). The reservoir contained 1 ml of precipitant buffer. Crystals were harvested after 3–4 days' growth and washed for 2–5 min in cryoprotection buffer containing 100 mM sodium citrate, pH 6.0, 100 mM magnesium acetate, 15.5% [w/v] polyethylene glycol 5000 monomethyl ether and 30% [v/v] ethylene glycol.

## 3.7. Data collection, structure determination and refinement

Diffraction data were collected from two cryo-protected crystals of $F_1$-ATPase from *F. nucleatum* at the Australian Synchrotron MX2 beamline [97], equipped with an ADSC Quantum 315r detector, and processed with XDS [98]. Because of radiation damage, four datasets from two crystals with the same space group and unit cell were merged with AIMLESS [99] during data reduction in CCP4 [100]. Molecular replacement was carried out with PHASER [101] using the $\alpha_3\beta_3$ subcomplex from the structure of the $F_1$-ATPase from *C. thermarum* containing the mutations Asp89Ala and Arg92Ala in the ε-subunit (5ik2) [52] with nucleotides and other ligands removed. Rigid body refinement and restrained refinement using non-crystallographic symmetry restraints were performed with REFMAC5 [102]. In between each refinement round with REFMAC5, parts of the structure were rebuilt manually with Coot [103]. The stereochemistry of the structure was assessed with MolProbity [104]. Electron density maps were calculated with FFT [100], and images of structures and electron density maps were prepared in PyMOL [105].

## 3.8. Rotation of the γ-subunit

Residues 23–33 of the *F. nucleatum* γ-subunit (and equivalent regions in other $F_1$-ATPases) interact with the C-terminal domains of the α- and β-subunits, and this segment acts as a rigid body uninfluenced by contacts in the crystal lattice between adjacent $F_1$-ATPase complexes. By contrast, residues 34–226 of the γ-subunit, and the associated δ- and ε-subunits lie outside the $\alpha_3\beta_3$-domain, where their positions may be subject to such influences. Therefore, the rotations of residues 23–33 of the γ-subunit in the various aligned structures were measured relative to the position of the same segment (residues 22–32) in the ground-state structure of azide-free bovine $F_1$-ATPase (2jdi) [12]. These measurements were made by aligning the structures via the crown domains at the N-termini of α- and β-subunits and then by calculating the centre of mass of residues 22–33 of the γ-subunit and determining the rotation angle around the pseudo-threefold axis of the α-subunits required to match its position with that of the equivalent segment in the bovine azide-free ground-state structure, which was taken as the reference point, set as 0° [10]. To calculate the rotation of the γ-subunit in molecule 2 with respect to that in molecule 1, the distances between the Cα of residues 95 and 105 (helix 2 in the γ-subunit) were measured in both molecules and then to their equivalent residues in the other model. These distances were extrapolated to a common origin giving the lengths of three sides of a triangle, allowing the angle to be calculated.

**Data accessibility.** The data available from the PDB accession code: 6q45.

**Authors' contributions.** Conceptualization: J.E.W and G.M.C. Methodology: J.P., Y.N., M.G.M., S.A.F., D.A., A.H. Investigation: J.P., Y.N., S.A.F., D.A., A.H. Formal analysis: J.P., Y.N., M.G.M. and A.G.W.L. Writing the original draft: J.E.W., J.P. and M.G.M. Writing review and editing: J.E.W., G.M.C., J.P., M.G.M., A.G.W.L. and S.A.F. Supervision: G.M.C. and J.E.W. Funding acquisition: G.M.C. and J.E.W.

**Competing interests.** The authors declare no competing interests.

**Funding.** This work was supported by a James Cook Fellowship from the Royal Society of New Zealand to G.M.C. and by the Medical Research Council, U.K. by grants MC_UU_00015/8 and MR/M009858/1 to J.E.W., and MC_U105184325 to A.G.W.L.

**Acknowledgements.** We thank the MX2 beamline staff at the Australian Synchrotron.

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
