## [Reviewer comments · Open Biology]

Review History

RSOB-19-0066.R0 (Original submission)

Review form: Reviewer 1

Recommendation

Accept with minor revision (please list in comments)

Are each of the following suitable for general readers?

- a) **Title**
Yes
- b) **Summary**
Yes
- c) **Introduction**
Yes

Is the length of the paper justified?

Yes

Should the paper be seen by a specialist statistical reviewer?

No

Is it clear how to make all supporting data available?

Yes

Is the supplementary material necessary; and if so is it adequate and clear?

Yes

Do you have any ethical concerns with this paper?

No

Comments to the Author

The structures of the F-type ATP synthases have proven to be more diverse than would have been anticipated by most, if not all, investigators. Perhaps even more unexpected has been the notion that this conserved enzyme may, on account of the species-specific structural difference, be a drug target. In addition, understanding the complexities of conformational changes and modes of inhibition by endogenous inhibitors can be enhanced by collecting multiple structures. It is in this context that this paper should be seen. It is scientifically a high standard contribution and the discussion of comparative points has to be presented even if it does become a major theme. A few minor points.

1. If read quickly line 65 can be taken to mean that ATP hydrolysis drives the flagella - .. hydrolysing ATP to generate....., or , in motile bacteria, to drive....
2. Is there any reference to 'vary over a wide range' (line 68)?
3. line 112 - this reviewer thought that thermostability is not so easy in general terms to understand as hoped in 1983 when ref 40 was written. Not a very important point here but could more recent references be given as well?
4. line 97 is there no delta subunit?

Review form: Reviewer 2

Recommendation

Accept with minor revision (please list in comments)

Are each of the following suitable for general readers?

- a) **Title**
Yes
- b) **Summary**
Yes
- c) **Introduction**
Yes

Is the length of the paper justified?

Yes

Should the paper be seen by a specialist statistical reviewer?

No

Is it clear how to make all supporting data available?

Yes

Is the supplementary material necessary; and if so is it adequate and clear?

Yes

Do you have any ethical concerns with this paper?

No

Comments to the Author

This manuscript characterizes the F1-ATPase from the pathogenic anaerobic bacterium *F. nucleatum*. The authors have done an excellent job of examining the catalytic activity of the enzyme by ensemble measurements as well as the structure determined by protein crystallography. Characterization of the structure and function of this protein complex may enable the design of new drugs against this pathogen. The work is important and will attract a broad readership.

During Lines 205 to 234, the authors discuss possible relationships between several F1-ATPase structures from bovine mitochondria to the rotary mechanism of this molecular motor as revealed by single-molecule rotation experiments. This is an important part of the manuscript that certainly increases the relevance of the work presented. However, the statements between lines 216 and 225 require a more thorough discussion and an adequate response to the queries below before the manuscript is suitable for publication:

1. A figure that shows the rotational alignment of the gamma and beta-empty subunits of the structures discussed is needed to increase confidence in the conclusions. These figures could be included in supplementary information. The authors claim structures 1H8E and 4YXW differ in rotation by 105 degrees, and conclude that these structures represent the ATP-binding dwell and the Phosphate release dwell. Aligning structures 4YXW and 4ASU would be of value to support their conclusion that the latter represents the catalytic dwell.
2. The 1H8E structure contains bound Mg-ADP-fluoroaluminate, which is a well-known transition state inhibitor of the ATP hydrolysis reaction. The authors must explain why this structure is not a good fit for the rotary position of the catalytic dwell when ATP hydrolysis occurs.
3. Although the structures show differences in rotational position in the globular domain of subunit gamma, the other end of this subunit has not rotated in any of the structures discussed due to a cluster of electrostatic interactions that exists between the "catch loop" of the empty beta subunit and subunit gamma. Greene and Frasch (2003) showed that mutating these highly conserved residues dramatically reduces catalytic function. The authors need to explain how their conclusions of rotary positions fit with the catch loop/gamma subunit electrostatic interactions.
4. The authors need to state the reasons why their structure/rotary position assignments are preferred to those reported by Suzuki et al. (2014) *Nature Chem Biol*, who originally characterized the rotary dwell positions of human F1-ATPase.
5. The authors also need to discuss why their structural assignment of 4ASU as the catalytic dwell is a better fit to available data than that of Nam, Pu and Karplus (2014) who concluded that 4ASU closely resembled the rotary position of the ATP-binding dwell.
6. Immediately after the discussion to relate bovine structures to rotary positions, lines 226 to 234

discusses the gamma subunit rotary positions of bacteria. For clarity the authors should state that single-molecule studies of *G. stearothermophilus* and *E. coli* F1 showed that Pi release occurs at the end of the catalytic dwell (Watanabe et al. (2010) Nature Chem Biol) and ADP release occurs about 25 degrees before the catalytic dwell (Martin et al. (2014) PNAS). These are approximately at the same rotary positions as human

Suzuki et al. (2014) Nature Chem Biol, but in reverse order.

7. As a minor point, it would be helpful to readers if the authors included the PDB reference numbers in the text when comparing so many different structures.

Decision letter (RSOB-19-0066.R0)

10-May-2019

Dear Professor Cook,

We are pleased to inform you that your manuscript RSOB-19-0066 entitled "Structure of F₁-ATPase from the Obligate Anaerobe *Fusobacterium nucleatum*" has been accepted by the Editor for publication in Open Biology. The reviewer(s) have recommended publication, but also suggest some minor revisions to your manuscript. Therefore, we invite you to respond to the reviewer(s)' comments and revise your manuscript.

Please submit the revised version of your manuscript within 7 days. If you do not think you will be able to meet this date please let us know immediately and we can extend this deadline for you.

- 1) A text file of the manuscript (doc, txt, rtf or tex), including the references, tables (including captions) and figure captions. Please remove any tracked changes from the text before submission. PDF files are not an accepted format for the "Main Document".
- 2) A separate electronic file of each figure (tiff, EPS or print-quality PDF preferred). The format should be produced directly from original creation package, or original software format. Please note that PowerPoint files are not accepted.
- 3) Electronic supplementary material: this should be contained in a separate file from the main

text and meet our ESM criteria (see <http://royalsocietypublishing.org/instructions-authors#question5>). All supplementary materials accompanying an accepted article will be treated as in their final form. They will be published alongside the paper on the journal website and posted on the online figshare repository. Files on figshare will be made available approximately one week before the accompanying article so that the supplementary material can be attributed a unique DOI.

Online supplementary material will also carry the title and description provided during submission, so please ensure these are accurate and informative. Note that the Royal Society will not edit or typeset supplementary material and it will be hosted as provided. Please ensure that the supplementary material includes the paper details (authors, title, journal name, article DOI). Your article DOI will be 10.1098/rsob.2016[last 4 digits of e.g. 10.1098/rsob.20160049].

4) A media summary: a short non-technical summary (up to 100 words) of the key findings/importance of your manuscript. Please try to write in simple English, avoid jargon, explain the importance of the topic, outline the main implications and describe why this topic is newsworthy.

Images

Data-Sharing

It is a condition of publication that data supporting your paper are made available. Data should be made available either in the electronic supplementary material or through an appropriate repository. Details of how to access data should be included in your paper. Please see <http://royalsocietypublishing.org/site/authors/policy.xhtml#question6> for more details.

Data accessibility section

Sincerely,

The Open Biology Team
<mailto:openbiology@royalsociety.org>

Reviewer(s)' Comments to Author:

Referee: 1

Comments to the Author(s)

The structures of the F-type ATP synthases have proven to be more diverse than would have been anticipated by most, if not all, investigators. Perhaps even more unexpected has been the notion

that this conserved enzyme may, on account of the species-specific structural difference, be a drug target. In addition, understanding the complexities of conformational changes and modes of inhibition by endogenous inhibitors can be enhanced by collecting multiple structures. It is in this context that this paper should be seen. It is scientifically a high standard contribution and the discussion of comparative points has to be presented even if it does become a major theme. A few minor points.

1. If read quickly line 65 can be taken to mean that ATP hydrolysis drives the flagella - .. hydrolysing ATP to generate....., or , in motile bacteria, to drive....
2. Is there any reference to 'vary over a wide range' (line 68)?
3. line 112 - this reviewer thought that thermostability is not so easy in general terms to understand as hoped in 1983 when ref 40 was written. Not a very important point here but could more recent references be given as well?
4. line 97 is there no delta subunit?

Referee: 2

Comments to the Author(s)

This manuscript characterizes the F1-ATPase from the pathogenic anaerobic bacterium *F. nucleatum*. The authors have done an excellent job of examining the catalytic activity of the enzyme by ensemble measurements as well as the structure determined by protein crystallography. Characterization of the structure and function of this protein complex may enable the design of new drugs against this pathogen. The work is important and will attract a broad readership.

During Lines 205 to 234, the authors discuss possible relationships between several F1-ATPase structures from bovine mitochondria to the rotary mechanism of this molecular motor as revealed by single-molecule rotation experiments. This is an important part of the manuscript that certainly increases the relevance of the work presented. However, the statements between lines 216 and 225 require a more thorough discussion and an adequate response to the queries below before the manuscript is suitable for publication:

1. A figure that shows the rotational alignment of the gamma and beta-empty subunits of the structures discussed is needed to increase confidence in the conclusions. These figures could be included in supplementary information. The authors claim structures 1H8E and 4YXW differ in rotation by 105 degrees, and conclude that these structures represent the ATP-binding dwell and the Phosphate release dwell. Aligning structures 4YXW and 4ASU would be of value to support their conclusion that the latter represents the catalytic dwell.
2. The 1H8E structure contains bound Mg-ADP-fluoroaluminate, which is a well-known transition state inhibitor of the ATP hydrolysis reaction. The authors must explain why this structure is not a good fit for the rotary position of the catalytic dwell when ATP hydrolysis occurs.
3. Although the structures show differences in rotational position in the globular domain of subunit gamma, the other end of this subunit has not rotated in any of the structures discussed due to a cluster of electrostatic interactions that exists between the "catch loop" of the empty beta subunit and subunit gamma. Greene and Frasch (2003) showed that mutating these highly conserved residues dramatically reduces catalytic function. The authors need to explain how their conclusions of rotary positions fit with the catch loop/gamma subunit electrostatic interactions.
4. The authors need to state the reasons why their structure/rotary position assignments are preferred to those reported by Suzuki et al. (2014) *Nature Chem Biol*, who originally characterized the rotary dwell positions of human F1-ATPase.

5. The authors also need to discuss why their structural assignment of 4ASU as the catalytic dwell is a better fit to available data than that of Nam, Pu and Karplus (2014) who concluded that 4ASU closely resembled the rotary position of the ATP-binding dwell.

6. Immediately after the discussion to relate bovine structures to rotary positions, lines 226 to 234 discusses the gamma subunit rotary positions of bacteria. For clarity the authors should state that single-molecule studies of *G. stearothermophilus* and *E. coli* F1 showed that Pi release occurs at the end of the catalytic dwell (Watanabe et al. (2010) Nature Chem Biol) and ADP release occurs about 25 degrees before the catalytic dwell (Martin et al. (2014) PNAS). These are approximately at the same rotary positions as human Suzuki et al. (2014) Nature Chem Biol, but in reverse order.

7. As a minor point, it would be helpful to readers if the authors included the PDB reference numbers in the text when comparing so many different structures.

Author's Response to Decision Letter for (RSOB-19-0066.R0)

See Appendix A.

Decision letter (RSOB-19-0066.R1)

30-May-2019

Dear Professor Cook

We are pleased to inform you that your manuscript entitled "Structure of F₁-ATPase from the Obligate Anaerobe *Fusobacterium nucleatum*" has been accepted by the Editor for publication in Open Biology.

Article processing charge

Please note that the article processing charge is immediately payable. A separate email will be sent out shortly to confirm the charge due. The preferred payment method is by credit card; however, other payment options are available.

Sincerely,

The Open Biology Team
mailto:openbiology@royalsociety.org

Appendix A

Manuscript RSOB-19-0066 by Petri et al.

We are grateful to the referees for their helpful comments. In the modified manuscript changes requested by the referees have been highlighted in yellow and we have responded to the points that they raise as follows.

Referee 1

1. If read quickly line 65 can be taken to mean that ATP hydrolysis drives the flagella - .. hydrolysing ATP to generate....., or , in motile bacteria, to drive....

In the revised manuscript, the sentence has been modified so as to avoid any possible ambiguity.

2. Is there any reference to 'vary over a wide range' (line 68)?

A reference has been provided; Dimroth & Cook, *Adv Microb Physiol*, 2004 see reference 4 in the revised list.

3. line 112 - this reviewer thought that thermostability is not so easy in general terms to understand as hoped in 1983 when ref 40 was written. Not a very important point here but could more recent references be given as well?

The general explanation of the basis for thermostabilization of proteins has not changed since the early 1980s. Proteins become thermostabilized by having additional energy in their structures, provided in many different ways: additional salt bridges, increased hydrophobic interactions, additional S-S bridges and so forth. They can also smooth off their external surface to prevent ingress of water molecules. We have provided an additional and more recent reference to the topic as requested; Kumar, Tsai & Nussinov, *Prot eng*, 2000.

4. line 97 is there no delta subunit?

There was no δ -subunit in the complex that was studied. This is now stated explicitly on line 98.

Referee 2

1. A figure that shows the rotational alignment of the gamma and beta-empty subunits of the structures discussed is needed to increase confidence in the conclusions. These figures could be included in supplementary information. The authors claim structures 1h8e and 4yxw differ in rotation by 105 degrees, and conclude that these structures represent the ATP-binding dwell and the Phosphate release dwell. Aligning structures 4yxw and 4asu would be of value to support their conclusion that the latter represents the catalytic dwell.

First, as requested by the referee, a new figure (Figure S8) showing γ - and β_E -subunits in several superimposed structures has been added to document the rotary positions of γ -subunits at catalytic and phosphate release dwells. This Figure needs to be considered with information already provided in Table S4 where the measured rotations of γ -subunits are summarized. Second, in the revised manuscript, we have also elaborated and explained explicitly in the text why the structure inhibited with thiophosphate (and rotationally similar structures without thiophosphate, eg the bovine high resolution ground state structure, 2jdi) represent the phosphate release dwell, and why structures inhibited by IF_1 , eg 2v7q (and rotationally similar structures without IF_1 , eg the phosphonate structure, 4asu) represent the catalytic dwell. All of these points have been published already (see Suzuki *et al*, *Nat Chem Biol*, 2014, Bason *et al*, *PNAS*, 2015), and the references were provided in the original manuscript, and additionally the points were summarized in Table S4. Third, we do not, as the referee asserts, conclude that the 1h8e structure is the ATP binding dwell, but we state clearly that it is possible that it could be, as it lies at a unique rotary position compared to all other structures and that position corresponds to neither phosphate release nor catalytic dwell.

2. The 1h8e structure contains bound Mg-ADP-fluoroaluminate, which is a well-known transition state inhibitor of the ATP hydrolysis reaction. The authors must explain why this structure is not a good fit for the rotary position of the catalytic dwell when ATP hydrolysis occurs.

First, Mg-ADP-fluoroaluminate is not a transition state inhibitor in ATP hydrolysis, it is a transition state analogue. Second, to repeat points made in response 1, in this structure, 1h8e, the γ -subunit is at a unique rotary position 15° ahead of the phosphate release dwell, and 45° before the position that we have identified as the catalytic dwell. This had already been documented in the original manuscript in Table S4. In addition, in the new Figure S8 where the various structures of γ - and associated β_E -subunits are superimposed, the unique rotational position of the γ - subunit in 1h8e is shown.

3. Although the structures show differences in rotational position in the globular domain of subunit gamma, the other end of this subunit has not rotated in any of the structures discussed due to a cluster of electrostatic interactions that exists between the “catch loop” of the empty beta subunit and subunit gamma. Greene and Frasch (2003) showed that mutating these highly conserved residues dramatically reduces catalytic function. The authors need to explain how their conclusions of rotary positions fit with the catch loop/gamma subunit electrostatic interactions.

All of our conclusions fit with the “catch” loop feature, and there is nothing to explain. We agree with the referee that above the “catch” loop all of the structures of γ -subunits superimpose and are essentially identical, and that it is below the “catch” loop that differences in the rotational states of γ -subunits are found. In the revised manuscript, we have provided a new Figure S7 to illustrate this point, and we have introduced the point into the text of the paper. The catch holds the γ -subunit to allow torsional energy to be stored somewhere below the catch, which once a critical point is reached releases the energy in a quantised manner to generate the rotational step or sub-step. The observed rotational dwells represented by the structures are presumably fixed stably by other as yet unidentified subtle changes in the structures of α - and β -subunits. At our current level of understanding, it is not possible to provide a more precise explanation as the information is not available.

4. The authors need to state the reasons why their structure/rotary position assignments are preferred to those reported by Suzuki et al. (2014) Nature Chem Biol, who originally characterized the rotary dwell positions of human F1-ATPase.

The numbers are arbitrary. As we are perfectly entitled to do, we set our highest resolution structure as the zero point and compared all other structures to it (this is a higher resolution version of the so-called “ground state” which was the first structure of an F₁-ATPase to be described). We have used these numbers in previously published papers without any adverse comments being made, and for the sake of consistency we prefer to continue to do so. In Fig. 6 of their paper, Suzuki *et al* show our structures on their rotary scale.

5. The authors also need to discuss why their structural assignment of 4asu as the catalytic dwell is a better fit to available data than that of Nam, Pu and Karplus (2014) who concluded that 4asu closely resembled the rotary position of the ATP-binding dwell.

The reasons for the assignment of 4asu as the catalytic dwell have been given above. Crucially, it should be noted that this assignment is based on two sets of complementary experimental data, our structures and the measured rotations of the γ -subunit, and “single molecule” experiments by Suzuki *et al* where rotary angles of the γ -subunit have been determined. The proposals of Nam *et al* are not based on experimental data but on simulations. Their claim that 4asu is at the ATP binding dwell is incorrect (the reasons have been given above), and, in addition, an inspection of Fig. S2B in the supplementary section of their paper shows that, contrary to what is stated in the paper, there are very significant structural differences between the positions of the β_{DP} -subunit and the γ -subunit based on their calculations and their positions in 4asu, resulting in steric clashes. They also assert that 1bmf represents the catalytic dwell whereas it clearly represents the phosphate release dwell; Suzuki *et al* demonstrated that IF₁ stopped rotation at the catalytic dwell. In the revised manuscript, we have added a comment saying that the conclusions based on the combination of determined structures and rotation experiments show that the proposals in Nam *et al.* are incorrect.

6. Immediately after the discussion to relate bovine structures to rotary positions, lines 226 to 234 discusses the gamma subunit rotary positions of bacteria. For clarity the authors should state that single-molecule studies of *G. stearothermophilus* and *E. coli* F1 showed that Pi release occurs at the end of the catalytic dwell (Watanabe et al. (2010) Nature Chem Biol) and ADP release occurs about 25 degrees before the catalytic dwell (Martin et al. (2014) PNAS). These are approximately at the same rotary positions as human Suzuki et al. (2014) Nature Chem Biol, but in reverse order.

Our failure to mention these observations in the original manuscript was a significant, albeit unintentional, omission. We thank the referee for pointing it out. In the revised manuscript, the observations have been mentioned and appropriate references have been included

7. As a minor point, it would be helpful to readers if the authors included the PDB reference numbers in the text when comparing so many different structures.

PDB references have been added throughout the text.